# Phase III, two arm, multi-centre, open label, parallel-group randomised designed clinical investigation of the use of a personalised early warning decision support system to predict and prevent acute exacerbations of chronic obstructive pulmonary disease: 'Predict & Prevent AECOPD' – study protocol

Dalbir Kaur [1], Rajnikant L Mehta,[2] Hugh Jarrett,[2] Sue Jowett,[3] Nicola K Gale,[4] Alice M Turner,[5,6] Monica Spiteri,[7] Neil Patel[8]

**Correspondence to**
Dalbir Kaur;
dalbir.kaur@warwick.ac.uk

## ABSTRACT

**Introduction** With 65 million cases globally, chronic obstructive pulmonary disease (COPD) is the fourth leading cause of death and imposes a heavy burden on patients' lives and healthcare resources worldwide. Around half of all patients with COPD have frequent (≥2 per year) acute exacerbations of COPD (AECOPD). Rapid readmissions are also common. Exacerbations impact significantly on COPD outcomes, causing significant lung function decline. Prompt exacerbation management optimises recovery and delays the time to the next acute episode.

**Methods/analysis** The Predict & Prevent AECOPD trial is a phase III, two arm, multi-centre, open label, parallel-group individually randomised clinical trial investigating the use of a personalised early warning decision support system (COPDPredict) to predict and prevent AECOPD. We aim to recruit 384 participants and randomise each individual in a 1:1 ratio to either standard self-management plans with rescue medication (RM) (control arm) or COPDPredict with RM (intervention arm). The trial will inform the future standard of care regarding management of exacerbations in COPD patients. The main outcome measure is to provide further validation, as compared with usual care, for the clinical effectiveness of COPDPredict to help guide and support COPD patients and their respective clinical teams in identifying exacerbations early, with an aim to reduce the total number of AECOPD-induced hospital admissions in the 12 months following each patient's randomisation.

**Ethics and dissemination** This study protocol is reported in accordance with the guidance set out in the Standard Protocol Items: Recommendations for Interventional Trials statement. Predict & Prevent AECOPD has obtained ethical approval in England (19/LO/1939). On completion of the trial and publication of results a lay findings summary will be disseminated to trial participants.

### STRENGTHS AND LIMITATIONS OF THIS STUDY

⇒ This is the first major trial of real-time observation, combined with personalised self-management in chronic obstructive pulmonary disease (COPD).
⇒ The minimisation algorithm addresses major associations of acute exacerbations of their COPD admission, such that these should not confound results.
⇒ There is a holistic exploration of deployment of a complex tool, because we have embedded qualitative work with staff and patients.
⇒ We have addressed digital exclusion by ensuring that funding is provided for mobile data, where patients do not have this, or WiFi, already.
⇒ Logistical issues to do with provision of mobile C-reactive protein measurement means we could only include a limited number of sites, and these do not reflect the whole of the UK.

**Trial registration number** NCT04136418.

## BACKGROUND

With 65 million cases globally, chronic obstructive pulmonary disease (COPD) is the fourth leading cause of death and imposes a heavy burden on patients' lives and healthcare resources worldwide.[1] UK death rates are currently double the European Union (EU) average with over 30 000 people dying yearly and annual COPD-related National Health Service (NHS) direct costs exceeding £800 million.[2] Patients can have acute exacerbations of their COPD (AECOPD),

which reduce quality of life, and lead to 140 000 emergency hospital admissions a year in the UK.[3] Indeed, AECOPD remain the second most common cause of emergency hospital admissions and one in three of these patients (sometimes more) will be readmitted within 3 months.[4][5] Hospitalisation itself carries a poor prognosis with an increased mortality risk.

Around half of all patients with COPD have frequent (≥2 per year) AECOPD.[6][7] Rapid readmissions are also common—the national COPD audit has shown that 43% of patients with COPD who were admitted are back in hospital within 90 days,[8] and up to 71% by 12 months.[9] Exacerbations impact significantly on COPD outcomes, causing significant lung function decline.[10] Prompt exacerbation management optimises recovery and delays the time to the next acute episode,[11] with National Institute for Health and Care Excellence COPD guidelines highlighting a time window (prodrome) between an initial exacerbation's symptoms/signs and subsequent hospitalisation. Within this prodrome, there is an opportunity to intervene. Current practice for COPD patients is that they are encouraged to recognise AECOPD, via standard self-management plans (SSMPs), and treat, using rescue medication (RM), but in many cases because of day-to-day variability in symptoms, the start of an exacerbation goes unrecognised and untreated. This inability/uncertainty to recognise and treat exacerbations in their early phase can lead to hospital admissions, which may drive long-term decline.

Pharmacological and non-pharmacological treatments are used to reduce COPD exacerbations,[12][13] and improve care, such as pulmonary rehabilitation (PR),[14] azithromycin,[15] inhaled bronchodilators and inhaled corticosteroids. Infection control measures during the pandemic reduced spread of other respiratory viruses, which in turn was associated with a 43% fall in admissions, with benefits maintained post-lockdown[16] though this is not currently part of international COPD guidelines.[17][18] SSMPs help patients but have not really shown significant impact on accident and emergency (A&E) visits or hospital admissions as demonstrated by a systematic review of self-management strategies for COPD.[19] Intuitively early recognition and treatment of AECOPD would reduce exacerbation severity and duration, and improve prognosis; evidence for this is limited but supportive.[20][21]

This study looks to address the problem by personalising and thus optimising effectiveness of AECOPD management using the CE marked and MHRA registered solution, COPDPredict.[22]

## METHODS
### Patient and public involvement
Throughout the design and delivery of the trial public and participant involvement (PPI) has been accessed. In particular independent members on the trial steering committee provide oversight and contributions to the trial, including design and standardisation of trial-specific SSMP. PPI involvement has also extended to the

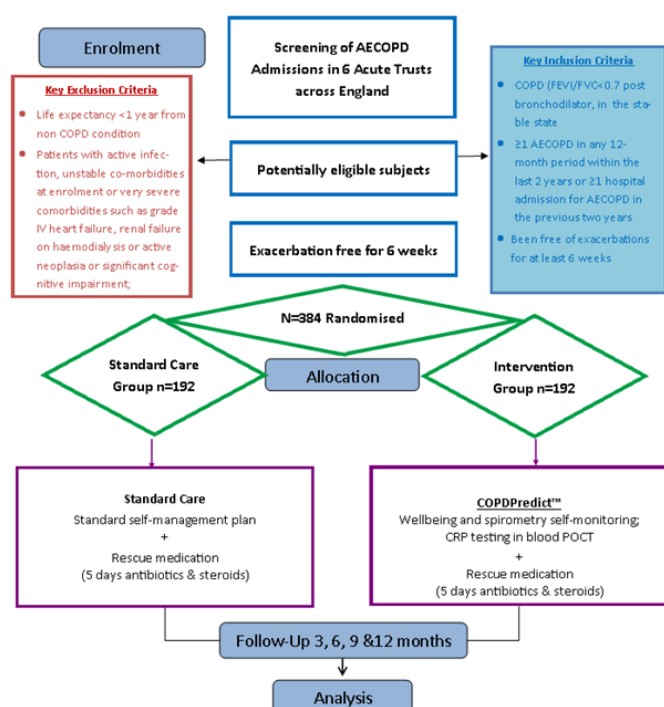

**Figure 1** Flow diagram of the Predict & Prevent trial. AECOPD, acute exacerbations of COPD; COPD, chronic obstructive pulmonary disease; CRP, C-reactive protein; FEV$_1$, forced expiratory volume; FVC, forced vital capacity; POC, point-of-care testing.

qualitative research where the topic guide was developed in conjunction with respiratory participant groups.

### Trial design
The Predict & Prevent AECOPD trial is a phase III, two arm, multi-centre, open label, parallel-group individually randomised clinical trial investigating the use of a personalised early warning decision support system (COPDPredict) to predict and prevent AECOPD. We aim to recruit 384 participants from NHS hospitals in the UK and randomise each individual in a 1:1 ratio to either SSMP with RM (control arm) or COPDPredict with RM (intervention arm) (figure 1). The start date for the trial was 30 January 2019 and the end date for the trial is 30 March 2023.

The main outcome measure is to provide further validation, as compared with usual care, for the clinical effectiveness of COPDPredict to help guide and support COPD patients and their respective clinical teams in identifying exacerbations early, with an aim to reduce the total number of AECOPD-induced hospital admissions in the 12 months following each patient's randomisation. The trial includes a health economic analysis examining health-related quality of life (HRQoL) using the validated questionnaire EQ-5D-5L and healthcare usage (as determined by a questionnaire that inquires as to hospitalisations, general practitioner (GP) attendances and medication usage). A qualitative substudy will assess both the end user experience and the healthcare provider experience, using

Normalisation Process Theory (NPT) to understand the implementation and integration of the new technology.[23]

## Eligibility criteria
### Inclusion criteria
► Clinically diagnosed COPD, confirmed by post-bronchodilator spirometry and defined as post-bronchodilator forced expiratory volume ($FEV_1$)/forced vital capacity <0.7 and <lower limit of normal, in the stable state.
► ≥1 AECOPD in any 12-month period within the last 2 years or ≥1 hospital admission for AECOPD in the previous 2 years.
► Exacerbation free for at least 6 weeks.
► An age of at least 18 years.
► Willing and able to comply with data collection to 12 months from randomisation.
► Ability to consent.
► Ability to use intervention as judged by the investigator at screening, on demonstration of the system to the patient.

### Exclusion criteria
► Life expectancy <12 months—assessment of the individual investigator.
► Co-enrolment into any clinical trials of investigative medicinal product.
► Patients with selected comorbidities, where these could impair use of the intervention (eg, dementia).

## OUTCOMES
### Primary outcome
Number of hospital admissions at 12 months postrandomisation where the primary reason for admission is AECOPD. This will be obtained from participant testimony and from centrally held (HES) records. Data from the two sources will be cross-referenced to remove double-counting. The total number of admissions will be the sum of unique admissions from the two sources of data.

### Secondary outcomes
#### Clinical outcomes
Over a 12-month period, following randomisation:

Total in participant days, number of visits to A&E, number of participants defined exacerbations, appropriate action on symptoms of AECOPD, as shown by the app, or participant testimony, HRQoL (COPD Assessment Test (CAT)) and, $FEV_1$ at 12 months postrandomisation (from spirometry obtained during study visit).

#### Economic outcomes
HRQoL at baseline and 3, 6, 9 and 12 months postrandomisation (EQ-5D-5L).

Healthcare usage at 3, 6, 9 and 12 months postrandomisation, as determined by a questionnaire that inquires on hospitalisations, GP attendances and medication usage.

#### Qualitative substudy
End user experience and provider experience

## CONTROL ARM
An SSMP alongside use of RM containing 5 days of antibiotic and steroid treatment.

## INTERVENTION
COPDPredict, consists of a patient-facing app (iOS/Android) and clinician-facing dashboard.[24] The early warning decision support system uses longitudinal remote monitoring of relevant subjective and objective data (symptoms, spirometry, biomarkers) in real-time via the app which is connected to CE-marked Bluetooth-enabled sensor peripherals. These data are used to construct COPD-relevant individual profiles such that artificial intelligence-driven algorithms can then identify changes in health status to provide timely individualised alerts to patients and clinicians and sign-posting to action plans for patients.

COPDPredict also provides information around COPD self-management, PR, inhaler technique and uses gamification to help with adherence. Patients have full access to their results and can also directly message their clinical team. COPDPredict may signpost a participant to a self-management action plan for exacerbation management but clinical team supervision and oversight will remain throughout, as per usual care.

The clinician-facing dashboard allows for 'real-time' case management with the ability to remotely monitor patients and facilitate interaction. Clinicians can choose to escalate treatments based on the results being transmitted by the patients. A crucial dimension to this advanced system is that blood biomarker measurements which inform an algorithm are incorporated to enhance accuracy. Blood-based point-of-care (POC) analysers are used with COPDPredict and these POC analysers have previously been validated against laboratory C-reactive protein measures.

## RECRUITMENT AND CONSENT
All patients must provide fully informed consent. A Participant Information Sheet which explains the different ways in which the participant may enter the trial is provided to participants to facilitate this process. Investigators or delegate(s) adequately explain the aim, trial intervention, anticipated benefits and potential hazards of taking part in the trial to the participant. They also stress that participation is voluntary and that the participant is free to refuse to take part and may withdraw from the trial at any time. Any trial specific procedures are completed after the participant has given informed consent. At each visit the participant's willingness to continue in the trial is ascertained and documented in the medical notes. Where new information becomes available which may affect the participants' decision to continue, participants will be given time to consider and if happy to continue will be re-consented. Re-consent will be documented in the medical notes.

## RANDOMISATION

After informed consent has been obtained and full participant eligibility confirmed (including exacerbation free for 6 weeks), the participant can be randomised into the trial (table 1). Randomisation Case Report Forms are provided to investigators and may be used to collate the necessary information prior to randomisation. Participants are randomised by computer/telephone at the level of the individual in a 1:1 ratio to either SSMP with RM (control arm) or COPDPredict with RM (intervention arm).

A minimisation algorithm will be used within the online randomisation system to ensure balance in the treatment allocation over centre, age (<60, ≥60 years) and severity of disease as per Global Initiative for Obstructive Lung Disease (GOLD).[1]

The participants are randomised to one of the following.

### Control arm

An SSMP alongside use of RM containing 5 days of antibiotic and steroid treatment.

### Intervention arm

Supported self-management using the COPDPredict app, involving personalised alerts to both patients and the clinical care team.

A 'random element' will be included in the minimisation algorithm, so that each participant has a probability, of being randomised to the opposite treatment that they would have otherwise received. Once randomised patients will also be asked to complete baseline quality of life booklet which contains both the EQ-5D-5L and CAT questionnaire.

## STATISTICAL ANALYSIS

### Sample size

The justification of the sample size is based on previous evidence[25] that had shown a mean estimate of 2.5 COPD admissions in the previous year in the control group. To detect a difference of 1 admission in the mean number of admissions between groups using the standard methods of difference between means and assuming SD of 2.6 with 90% power and a type I error rate of 5% (two-sided), 144 participants per group will need to be randomised, 288 in total. Assuming and adjusting for a 25% loss to follow-up/drop-out rate, 384 participants will need to be recruited.

### Analysis of outcome measures

The primary comparison groups will be composed of those randomised to the use of COPDPredict system and those randomised to the SSMP. All analyses will be based on the intention to treat principle,[26] with all patients analysed in the treatment groups to which they were allocated irrespective of compliance with the randomised allocated treatment, and all patients will be included in the analyses. For all outcomes, summary statistics (eg, mean differences, relative risks) will be reported and 95% CIs will be constructed where appropriate. A two-sided p value of <0.05 will be considered statistically significant.

### Primary outcome measure

The primary analysis for this study will be to compare rates of hospital admission between the treatment groups (COPDPredict vs usual care) over 12 months, following randomisation, where the primary reason for admission is an AECOPD event. These hospital admissions per person will be analysed using Poisson regression models adjusting for treatment group and minimisation variables. If there is over dispersion a negative binomial regression model adjusting for treatment group and the same minimisation variables will be taken into consideration. Point estimates (incidence rates) will be provided and accompanied with 95% CIs and respective p values.

### Secondary outcome measures

Recorded observations taken over a 12-month period, following randomisation.

#### Total in-hospital days and FEV$_1$

These variables will be summarised using basic descriptive statistics (mean, SD). We may also consider a mixed linear regression model, to estimate differences between the intervention group supported with 95% CI adjusting for baseline variables and minimisation variables (centre being a random effects variable).

#### HRQoL questionnaires

EQ5D-5L and CAT will be converted into scores and analysed using a mixed linear regression model, adjusting for the intervention group, baseline and minimisation variables as before.

#### Number of A&E visits, participant defined exacerbations and healthcare usage

Where the participant experienced an episode, the data during the 12 months from randomisation will be analysed using mixed effects log-binomial regression techniques with presentation of relative risk and 95% CIs. Furthermore, sensitivity analysis based on patients' experience of multiple visits to A&E and participant defined exacerbation will be calculated using Poisson regression techniques and relative risk supported with 95% CI.

#### Symptom control marker association to clinical decision

The diagnostic accuracy of COPDPredict and Usual care will be evaluated by calculating sensitivity, specificity, positive and negative predictive values, and area under curve/receiver operating characteristics together with 95% CI.

### Subgroup analyses

Subgroup analyses will be limited to the same variables used in the minimisation algorithm, apart from centre. Tests for statistical heterogeneity (eg, by including the treatment group by subgroup interaction parameter in the statistical model) will be performed prior to any examination of effect estimate within subgroups. The results of

**Table 1** Schedule of assessments

| Visit | Hospital discharge/initial appointment | Personal baseline day 1 after 6 weeks exacerbation free | Personal baseline day 7 (±2 days) | End of personal baseline period | Approximately four times during any exacerbation (home visit or clinic) | Weekly until week 52 | Week 13 telephone follow-up ±1 week | Week 26 hospital/telephone follow-up ±1 week | Week 39 telephone follow-up ±1 week | Week 52 telephone/hospital follow-up ±2weeks | Patients daily well-being completion post 2-week baseline period |
|---|---|---|---|---|---|---|---|---|---|---|---|
| Eligibility check | X | | | | | | | | | | |
| Valid informed consent | X | | | | | | | | | | |
| Randomisation | | X | | | | | | | | | |
| Height and weight | X | | | | | | | | | X | |
| Concomitant medication | X | X | | X | | | | X | X | X | |
| Baseline medical history taken | X | | | | | | | | | | |
| Request to GP for rescue medication | X | | | | | | (X) | (X) | (X) | | |
| Assessment of (S)AEs | X | | | | (X) | | X | X | X | X | |
| Questionnaires (QoL and health usage) | X | | | | | | X | X | X | X | |
| Spirometry | X | | | | | | | X | | X | |
| Intervention only below this line | | | | | | Intervention only below this line | | | | | |
| Provision of equipment and participant training | | X | | | | | | | | | |
| Spirometry | | X | X | X | (X) | X | | | | | |
| Symptom control markers | | X | X | X | (X) | X | X | X | X | X | |
| Point-of-care blood test | | X | | X | (X) | | | | | | |
| Clinical histories | | X | | X | | | X | X | X | X | |
| Well-being (app) | | | | | | | | | | | X |

'(X)' denotes only when required. If rescue medication has not been used, fresh prescription requests will not be necessary.
During the baseline period and any subsequent exacerbations patients will be entering well-being data on the app which will be related to lifestyle choices and the end-user experience. These data will not form part of the formal analysis of quality of life.
GP, general practitioner; QoL, quality of life; (S)AEs, (serious) adverse events.

subgroup analyses will be treated with caution and will be used for the purposes of hypothesis generation only.

## Missing data and sensitivity analyses

Participants with missing primary outcome data will not be included in the primary analysis in the first instance. This presents a risk of bias, and sensitivity analyses will be undertaken to assess the possible impact. This will consist of simulating the missing response using a multiple imputation approach.[27] Parameters used to simulate the missing response will include the minimisation variables, intervention group and previous response. Full details are included in the statistical analysis plan.

## Adverse events and serious adverse events

Patients with COPD can have high disease burden, such that the trial population will be older, with comorbidities such as osteoporosis,[28] cardiovascular disease or raised cardiovascular risk,[29] along with associated complications and symptoms including abnormal lab results. A relatively high number of adverse events (AEs) are anticipated as a result of the patients' existing medical history.[30] However, this study is examining the role of increased participant surveillance and therefore few, if any, foreseeable risks of direct harm associated with the study intervention are anticipated. AE reporting will therefore be limited to those events which are required for trial monitoring or outcome assessment.

All events which meet the definition of serious must be recorded in the participant notes, but for trial purposes these following events do not require reporting on the serious adverse event (SAE) form. Such events are 'safety reporting exempt'.

► Planned hospitalisation for a pre-existing condition, or a procedure required by the trial protocol without serious deterioration in health, is not considered a SAE.
► An overnight stay in hospital that is due to transportation, organisational or accommodation problems, and without medical background SAEs requiring expedited reporting.

Any other events that fulfil the usual definition of an SAE will be reported immediately.

Any death occurring during the trial protocol defined follow-up period (12 months) must be reported as an SAE within 24 hours of the local investigator becoming aware of the event.

## HEALTH ECONOMICS
### Within-trial analysis

The economic evaluation alongside the trial will take the form of an incremental cost-effectiveness analysis to estimate cost per hospital admission avoided and incremental cost-utility analysis to estimate cost per quality-adjusted life year (QALY) gained. Base-case analyses will be from a healthcare perspective over 12-month follow-up using individual participant level data on costs and outcomes from the clinical trial. Additional analyses will be undertaken from a broader societal perspective. Resource use

data will be collected by questionnaire, capturing information on hospitalisations for exacerbations, COPD-related primary and secondary care attendances and medication usage. Data will also be collected during the trial on resource use implications of the intervention. Information on participant incurred costs, time off work and impact on activities is collected within the economic questionnaire. QALYs will be calculated from EQ-5D-5L responses. Sensitivity analysis will be undertaken and cost-effectiveness planes and cost-effectiveness acceptability curves presented.

### Model-based analysis

A decision model will be constructed to extrapolate beyond trial results over 5 years, with costs and benefits discounted at 3.5%. The model will be the first decision analysis to use GOLD stages ABCD[1] and will consider the ability of the COPDPredict supported self-management to impact movements between GOLD ABCD stages, but also the number of exacerbations within those stages. Costs associated with delivering COPDPredict, exacerbations, and routine care will be obtained from the trial. Quality of life scores on each intervention will also be obtained from trial data using EQ-5D-5L. Longer-term costs, quality of life scores and transitions between stages, and number of exacerbations will be obtained from analysis of ABCD cohorts from within the BLISS[2] study, a prospective cohort study with a 3-year follow-up period. The model will estimate cost per QALY gained of COPDPredict supported self-management versus standard self-management from a healthcare perspective. Probabilistic sensitivity analysis will be undertaken to simultaneously incorporate all parameter uncertainty. Cost-effectiveness planes and cost-effectiveness acceptability curves will be presented to show the probability the intervention is cost-effective at different cost/QALY thresholds. A value of information analysis will assess the value of further trials to reduce parameter uncertainty and identify parameters which would be most valuable in reducing that uncertainty.[31]

## QUALITATIVE SUBSTUDY

In order to frame this study, we draw on NPT,[32] which proposes four constructs essential to effect implementation of new technologies: *coherence* (sense making work to understand the possibilities of a new technology), *cognitive participation* (relational work as new technology begins to be used), *collective action* (operational work enacted to make a technology function in context) and *reflexive monitoring* (the appraisal of work that people do to assess how the new technology affects them and others). Participants will include healthcare professionals, patients and wider policy stakeholders and will be purposively sampled to maximise diversity. A topic guide developed by drawing on existing literature and theory will focus on attitudes to and practices around self-management of COPD and the use and implementation of digital

technologies. Interviews will be audio recorded and transcribed verbatim, before subjecting the data to a two-stage approach to analysis:[33] first, a descriptive framework analysis[34] followed by mapping onto the NPT constructs. We anticipate conducting approximately 30 interviews but this will be informed by the data and analysis.[35]

## ETHICS AND DISSEMINATION

This study protocol is reported in accordance with the guidance set out in the Standard Protocol Items: Recommendations for Interventional Trials statement. Predict & Prevent AECOPD has obtained ethical approval in England (19/LO/1939). On completion of the trial and publication of results a lay findings summary will be disseminated to trial participants.

## TRIAL OVERSIGHT

Trial oversight will be provided by the Trial Management Group, an independent Data Monitoring Committee (DMC) and independent Trial Steering Committee. Interim analyses of major outcome measures and safety data will be conducted and provided in strict confidence to the DMC.

## POTENTIAL IMPACT

Predict & Prevent AECOPD is a pragmatic, national, multi-centre, Randomised Control Trial (RCT) which aims to provide robust evidence around the clinical and cost-effectiveness of COPDPredict by analysing the number of patients admitted to hospital in a 12-month period postrandomisation. It is hoped the outcome of this trial could be used to provide robust evidence to the management of patients with COPD and their exacerbations in the future.

**Author affiliations**
[1]Warwick Clinical Trials Unit (BWCTU), Warwick Medical School University of Warwick Coventry, Coventry, UK
[2]Birmingham Clinical Trials Unit (BCTU), Institute of Applied Health Research, College of Medical and Dental Sciences, University of Birmingham, Birmingham, UK
[3]Health Economics Unit, Institute of Applied Health Research, College of Medical and Dental Sciences, University of Birmingham, Birmingham, UK
[4]Health Services Management Centre, School of Social Policy Director of Postgraduate Research, College of Social Sciences, University of Birmingham, Birmingham, UK
[5]Institute of Applied Health Research, College of Medical and Dental Sciences, University of Birmingham, Birmingham, UK
[6]Respiratory Medicine, Institute for Applied Health Research, University of Birmingham, Birmingham, UK
[7]Respiratory Research, Academic Research Unit, Royal Stoke University Hospital, University Hospitals of North Midlands NHS Trust, Staffordshire, UK
[8]NEPESMO Ltd, Manchester, UK

**Contributors** All authors have made substantial contributions to the conception or design of the work, or the acquisition, analysis or interpretation of data. DK, AMT, HJ, RLM, SJ, NKG, NP and MS drafted the manuscript and revised it critically for important intellectual content. All authors approved final version and agreed to be accountable for all aspects of the work in ensuring that questions related to the accuracy or integrity of any part of the work are appropriately investigated and resolved.

**Funding** This report is independent research funded by the National Institute for Health Research (Invention for Innovation (i4i), A Personalised Early Warning Decision Support System with novel Saliva Bio-Profiling to Predict and Prevent Acute Exacerbations of Chronic Obstructive Pulmonary Disease – 'Predict&prevent AECOPD, NIHR200002). The views expressed are those of the author(s) and not necessarily those of the NHSNational Health Service, the NIHR or the Department of Health.

**Competing interests** The CI of the trial (AMT) does not have any relevant direct financial disclosures, nor do members of the Trial Management Group (TMG) who are authors of this paper, with the exception of NP who is a founder, director and shareholder of NEPeSMO, who developed the intervention. In addition, MS is CI of the overall NIHR-funded project and is a founder, director and shareholder of NEPeSMO. NEPeSMO is a start-up company from the University Hospitals of North Midlands NHS Trust, owns all intellectual property rights of COPDPredict and is a project collaborator on the grant. AMT has grants from pharmaceutical companies working in the area of COPD (Chiesi, AstraZeneca) and has conducted advisory work for such (Boehringer, CSL Behring) but not in the area of medical devices or admission prevention. Neither has she worked for, or received monies from, any company working on admission prevention in the last 3 years.

**Patient and public involvement** Patients and/or the public were involved in the design, or conduct, or reporting, or dissemination plans of this research. Refer to the Methods section for further details.

**Patient consent for publication** Not applicable.

**Provenance and peer review** Not commissioned; externally peer reviewed.

**ORCID iD**
Dalbir Kaur http://orcid.org/0000-0003-2722-3848

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
