## [Reviewer comments · BMJ Open]

ARTICLE DETAILS

TITLE (PROVISIONAL)	A phase III, 2 arm, multi-centre, open label, parallel-group randomised designed clinical investigation of the use of a personalised early warning decision support system to predict and prevent acute exacerbations of Chronic Obstructive Pulmonary Disease - 'Predict & Prevent AECOPD' -Study Protocol
AUTHORS	Kaur, Dalbir; Mehta, Rajnikant; Jarrett, Hugh; Jowett, Sue; Gale, Nicola K.; Turner, Alice; Spiteri, Monica; Patel, Neil

VERSION 1 – REVIEW

REVIEWER	Sivapalan, Pradeesh Herlev and Gentofte Hospital, Herlev Gentofte Hospital
REVIEW RETURNED	18-Apr-2022

GENERAL COMMENTS	It is an interesting project and important hypothesis to investigate interventions that can prevent COPD exacerbations. However, I have some questions/concerns about the protocol: 1) Regarding exclusion criteria, I think it should be defined more clearly (a) how assesses Life expectancy <12 months. Is it an assessment the individual investigator takes? (b) Patients with selected co-morbidities, where these could impair use of the intervention (eg dementia). These co-morbidities should be mentioned in the protocol so that the investigator is not in doubt (c) Maybe consider whether asthma patients should be excluded? 2) Primary Outcome Number of hospital admissions at 12 months post randomisation where the primary reason for admission is AECOPD. Is this the same as "frequency of hospital-requiring COPD exacerbations within a year" (3) This will be obtained from participant testimony and from centrally held (HES)records. Data from the 2 sources will be cross-referenced to remove double-counting. The total number of admissions will be the sum of unique admissions from the 2 sources of data how valid is it to use patient testimony? Often, COPD patients can be very ill, and have difficulty remembering admissions. Why not subtract all admissions from registries?
--

REVIEWER	Bourke, Stephen North Tyneside General Hospital, Respiratory Medicine
REVIEW RETURNED	21-Apr-2022

GENERAL COMMENTS	The protocol is clearly presented, and the selection criteria are broad, enhancing recruitment to the trial and the generalisability of the results. The trial addresses an area of urgent clinical need in a vulnerable population. I have made a few minor comments below. Abstract: please define all abbreviations when first used – standard self-management plans (SSMP), and rescue medication (RM). Page 5, paragraph 1: 90 day readmission rates have risen further beyond 1 in 3, in tandem with falling in-hospital mortality and length of stay The NACAP data (43%) is already cited in the next paragraph. Page 5, paragraph 3: vaccination and prophylactic azithromycin are worthy of mention in relation to exacerbation prevention. Infection control measures during the pandemic reduced spread of other respiratory viruses, which in turn was associated with a 43% fall in admissions, with benefits maintained post-lockdown - see doi.org/10.3390/medicina58010066 – though this is not currently part of (inter)national COPD guidelines. Page 7-8, outcomes: community treated AECOPD are hard to capture and verify, thus the focus on hospitalised events for the primary outcome is appropriate. Total hospital days are also captured as a secondary outcome - it is plausible that prompt treatment may reduce the length of stay even when hospitalisation is not averted. There may also be an impact on survival. An alternative combined outcome measure to consider capturing admissions, length of stay and mortality is: “Days alive out of hospital over 1 year post randomisation.” P 10: minimisation criteria: tools have been developed to predicted readmission risk and offer a means of combining a number of indices into a summary risk score. GOLD – helpful to specify as GOLD stage (1-4) or GOLD group (A-D) for clarity.
--

VERSION 1 – AUTHOR RESPONSE

Reviewer: 1

Dr. Pradeesh Sivapalan, Herlev and Gentofte Hospital Comments to the Author:

It is an interesting project and important hypothesis to investigate interventions that can prevent COPD exacerbations.

However, I have some questions/concerns about the protocol:

1) Regarding exclusion criteria, I think it should be defined more clearly

(a) how assesses Life expectancy <12 months. Is it an assessment the individual investigator takes? **Yes, added to the manuscript**

(b) Patients with selected co-morbidities, where these could impair use of the intervention (eg dementia). These co-morbidities should be mentioned in the protocol so that the investigator is not in doubt. **We have listed some examples of key co-morbidities for investigators, however given that comorbidity is common in COPD, it would be difficult to provide a list which covered all eventualities.**

(c) Maybe consider whether asthma patients should be excluded? **We decided this would be inappropriate because of the overlap seen between asthma and COPD diagnoses, in particular in primary care, where large numbers of UK patients are observed to be coded for both. In order to keep**

our trial pragmatic and generalisable to the COPD population seen in real life, no specific asthma exclusion is required.

2) Primary Outcome

Number of hospital admissions at 12 months post randomisation where the primary reason for admission is AECOPD.

Is this the same as "frequency of hospital-requiring COPD exacerbations within a year" **Yes**

(3) This will be obtained from participant testimony and from centrally held (HES) records. Data from the 2 sources will be cross-referenced to remove double-counting. **The total number of admissions will be the sum of unique admissions from the 2 sources of data**

how valid is it to use patient testimony? Often, COPD patients can be very ill, and have difficulty remembering admissions. Why not subtract all admissions from registries? **In this case patient testimony is backed up by the GP and hospital record, hence patient testimony is not the only source of information. HES extracts alone were not felt to be appropriate because of delays in obtaining data and potential coding errors, which could miss AECOPD events (eg misclassification as pneumonia)**

Reviewer: 2

Dr. Stephen Bourke, North Tyneside General Hospital, Newcastle University Comments to the Author: The protocol is clearly presented, and the selection criteria are broad, enhancing recruitment to the trial and the generalisability of the results. The trial addresses an area of urgent clinical need in a vulnerable population.

I have made a few minor comments below.

Abstract: please define all abbreviations when first used – standard self-management plans (SSMP), and rescue medication (RM). Address abbreviations **-Done**

Page 5, paragraph 1: 90 day readmission rates have risen further beyond 1 in 3, in tandem with falling in-hospital mortality and length of stay The NACAP data (43%) is already cited in the next paragraph. **Reworded**

Update frequency to latest from NACAP or find a more recent reference supporting the rate we quote Page 5, paragraph 3: vaccination and prophylactic azithromycin are worthy of mention in relation to exacerbation prevention.

Infection control measures during the pandemic reduced spread of other respiratory viruses, which in turn was associated with a 43% fall in admissions, with benefits maintained post-lockdown - see doi.org/10.3390/medicina58010066 – though this is not currently part of (inter)national COPD guidelines. **Reworded and citation added**

Page 7-8, outcomes: community treated AECOPD are hard to capture and verify, thus the focus on hospitalised events for the primary outcome is appropriate.

Total hospital days are also captured as a secondary outcome - it is plausible that prompt treatment may reduce the length of stay even when hospitalisation is not averted. There may also be an impact on survival. An alternative combined outcome measure to consider capturing admissions, length of stay and mortality is: "Days alive out of hospital over 1 year post randomisation." **This is a good idea and could be added to our exploratory outcomes P 10: minimisation criteria: tools have been developed to predicted readmission risk and offer a means of combining a number of indices into a**

summary risk score. GOLD – helpful to specify as GOLD stage (1-4) or GOLD group (A-D) for clarity.
Add the ABCD element

VERSION 2 – REVIEW

REVIEWER	Bourke, Stephen North Tyneside General Hospital, Respiratory Medicine
REVIEW RETURNED	15-Aug-2022
GENERAL COMMENTS	The authors have addressed the minor concerns raised. Randomisation by minimisation: the investigators plan to randomise a proportion of participants by simple randomisation, which is appropriate - worth including the proportion selected by the trial statistician. Secondary outcomes: Total inpatient days has been retained; deaths during the follow up period will reduce the period at risk. How will this be handled statistically? Total days alive out of hospital over one year offers one approach to this issue.